# In Vitro Activity of the Arylaminoartemisinin GC012 against *Helicobacter pylori* and Its Effects on Biofilm

**DOI:** 10.3390/pathogens11070740

**Published:** 2022-06-29

**Authors:** Francesca Sisto, Simone Carradori, Sarah D’Alessandro, Nadia Santo, Norma Lattuada, Richard K. Haynes, Donatella Taramelli, Rossella Grande

**Affiliations:** 1Department of Biomedical, Surgical and Dental Sciences, University of Milan, Via Pascal 36, 20133 Milan, Italy; 2Department of Pharmacy, “G. d’Annunzio” University of Chieti-Pescara, Via dei Vestini 31, 66100 Chieti, Italy; simone.carradori@unich.it (S.C.); rossella.grande@unich.it (R.G.); 3Department of Pharmacological and Biomolecular Sciences, University of Milan, Via Pascal 36, 20133 Milan, Italy; sarah.dalessandro@unimi.it (S.D.); donatella.taramelli@unimi.it (D.T.); 4UNITECH—University Technological Platform, University of Milan, Via Golgi 19, 20133 Milan, Italy; nadia.santo@unimi.it (N.S.); norma.lattuada@unimi.it (N.L.); 5Centre of Excellence for Pharmaceutical Sciences, Faculty of Health Sciences, North-West University, Potchefstroom 2520, South Africa; haynes@ust.hk

**Keywords:** *Helicobacter pylori*, biofilm, MIC, MBIC, MBEC, dihydroartemisinin, amino-artemisinin, synergism, antibiotics, FICI

## Abstract

This study evaluated the in vitro activity of the arylaminoartemisinin GC012, readily obtained from dihydroartemisinin (DHA), against clinical strains of *Helicobacter pylori* (*H. pylori*) with different antibiotic susceptibilities in the planktonic and sessile state. The activity was assessed in terms of bacteriostatic and bactericidal potential. The minimum inhibitory concentration (MIC) and minimum bactericidal concentration (MBC) were determined by the broth microdilution method. After treatment with GC012, all bacterial strains showed significantly lower MIC and MBC values compared to those of DHA. The effect of combination of GC012 with antibiotics was examined using the checkerboard method. GC012 displayed synergistic interactions with metronidazole, clarithromycin, and amoxicillin in all the strains. The antibiofilm activity was evaluated via crystal violet staining, AlamarBlue^®^ assay, colony-forming unit count, and fluorescence microscopy. At ½ MIC and ¼ MIC concentration, both GC012 and DHA inhibited biofilm formation, but only GC012 showed a minimal biofilm eradication concentration (MBEC) on mature biofilm. Furthermore, both compounds induced structural changes in the bacterial membrane, as observed by transmission electron microscopy (TEM) and scanning electron microscopy (SEM). It is thereby demonstrated that GC012 has the potential to be efficacious against *H. pylori* infection.

## 1. Introduction

*H. pylori* is a spiral-shaped, Gram-negative (Gram-), microaerophilic bacterium, which infects about 50% of the world’s population, with higher prevalence in developing countries [1]. In 1994, *H. pylori* was classified as type one carcinogen by the World Health Organization International Agency for Research on Cancer (IARC) [2]. It is considered a major agent responsible for the development of gastric ulcer, gastritis, and MALT lymphoma and, moreover, is a major risk factor for gastric carcinoma, one of the leading types of cancer [3].

So, the removal of *H. pylori* using antibiotics contributes to the prevention of stomach cancer in humans [4]. Multiple drug resistance (MDR), one of the most important challenges of our century, greatly affects the efficacy of therapeutic regimens used for treatment of *H. pylori* infection [5,6,7]. Furthermore, the alarming presence of *H. pylori* strains with a MDR profile, is increasing [8]. The regimen currently used as first-line treatment is quadruple therapy (proton pump inhibitor (PPI), tetracycline, metronidazole, and bismuth); in case of treatment failure, a second line quinolone-containing therapy (fluoroquinolone, amoxicillin, and PPI or levofloxacin, amoxicillin, PPI, and bismuth) is suggested for eradication [9,10,11]. In addition, clinical studies involving novel potassium-competitive acid blockers (P-CABs instead of proton pump inhibitor—PPIs) have been carried out [12] as the first-line treatment of multidrug-resistant *H. pylori* strains, particularly in areas of high clarithromycin resistance. The prevalence of *H. pylori* infections varies widely according to geographic area, age, race, and socioeconomic status [13]. The presence of bacterial drug resistance also varies in different geographic areas, and generally follows the level of drug use in a particular country [14].

Furthermore, it has been reported that *H. pylori* exhibits a high frequency of mutation and genetic recombination, which certainly contributes to the rapid emerging antibiotic resistance of *H. pylori* in many areas of the world [15,16]. The Review on Antimicrobial Resistance, commissioned by the UK Government, reported that antimicrobial resistance (AMR) will kill 10 million people per year by 2050 [17]. In 2017, the WHO produced a global priority pathogens list (global PPL) of antibiotic-resistant bacteria, in which clarithromycin-resistant *H. pylori* was assigned high priority for antibiotic research and development [18].

In addition, the treatment failure can be also due to poor compliance of the host immune system or to biofilm production that may contribute to recurrence of infection [19]. Biofilms are microbial communities embedded within a self-produced extracellular polymeric substance (EPS) containing proteomannanas, lipopolysaccharide (LPS)-related structures, extracellular DNA (eDNA), proteins, and outer membrane vesicles. This allows bacteria to both escape antimicrobial therapy and clearance from the immune system, and to persist in the host [20,21,22]. Furthermore, bacteria in biofilms are more resistant to antibiotics compared to free-living bacteria [23]. Thus, any compound with an antimicrobial effect that also inhibits biofilm formation or acts on bacteria biofilm would have the best potential for therapeutic use as a drug [24]. We focus here on derivatives of the peroxidic antimalarial drug artemisinin (Figure 1), originally isolated in 1972 from the traditional medicinal herb *Artemisia Annua* [25] and used for the treatment of malaria [26]. The derivatives dihydroartemisinin (DHA, Figure 1), artemether, and artesunate are now used worldwide in combination with other antimalarial drugs for front line treatment of malaria [27,28,29].

Artemisinins are very active against the malaria parasite, principally *Plasmodium falciparum* (*P. falciparum*), and can be used in relatively brief (*ca* 3 day) treatment regimens [30]. Artemisinin has also been converted into a large number of other derivatives that are shown to possess activity against both apicomplexan and non-apicomplexan parasites [31], bacteria, viruses [32,33], and also against cancer [34].

Here, we focus on amino-artemisinins that have enhanced efficacy against the blood stages of the malaria parasite [35,36]. Of these, the best known is artemisone (Figure 1), that has been used in a clinical trial in patient with non-severe malaria, wherein it elicited cure at one-third the dose level of the comparator drug artesunate. The potent biological activity is ascribed to the properties of the amino group at C-10 of the artemisinin nucleus, improved pharmacokinetics, and generation of active metabolites with relatively long half-lives [37].

In our previous study, an antimicrobial effect of DHA and artemisone against *H. pylori* has been reported [38]. Here, we examine the arylaminartemisinin GC012, which is obtained via a notably straightforward phase-transfer reaction from DHA and 4-fluoroaniline [39]. It is potently active against *P. falciparum* in vitro and is also active against the apicomplexan parasite *Neospora caninum* [40].

The aim of this study was to extend the observations by comparing the activities of GC012 and DHA against *H. pylori* under planktonic and biofilm growth conditions, and to examine the effects of both compounds on bacterial cells via TEM and SEM analysis.

## 2. Results

### 2.1. Antimicrobial Activity of GC012

The antimicrobial activity of GC012 was tested on clinical isolates, as well as the reference strain *H. pylori* ATCC 43504. As shown in Table 1, GC012 appeared to have stronger activity than DHA. Furthermore, the MIC_50_ value of GC012 was 16-fold lower than that of metronidazole (MNZ) and clarithromycin (CLR), and 2-fold higher than amoxicillin (AMX). The bactericidal activity was also assessed using the microdilution method. By considering the MBC range and the mean MBC_50_ values, GC012 was shown to be 32-fold more potent than DHA and MNZ, 8-fold more potent than CLR, and 4-fold less potent than AMX. The activity of both artemisinin derivatives was not related to the antimicrobial susceptibility of the strains.

In order to verify the specificity of GC012 against *H. pylori*, a panel of twelve bacterial strains and two yeast strains was also investigated. As shown in Table 2, GC012 was found to be inactive against both bacteria and yeast strains.

### 2.2. Effect of pH on the Stability of GC012

Because of the low pH (1.5 to 3.5) in the stomach, any molecule for the treatment of H. pylori infection must be active under these conditions. Furthermore, it is important to consider that in the stomach, a pH gradient also exists [41]. In fact, in the mucus layer, the pH increases to ~6, reaching a value of ~7 in the proximity of the gastric epithelium where H. pylori establishes the infection. For this reason, we investigated the stability of the test compounds at different pH values, in terms of antimicrobial activity (MIC and MBC), against H. pylori ATCC 43504 and five clinical isolates with different susceptibilities to MNZ and CLR (Table 3). Interestingly, the variation of pH did not alter the activity of GC012 against all the strains tested. This is different from DHA, as we previously demonstrated [38]. On the basis of these results, it can be concluded that bactericidal activity of GC012 is not altered by the pH. 

### 2.3. Killing Kinetics

A kinetic study was performed against *H. pylori* ATCC 43504 (resistant to MNZ), and two clinical isolates with different antibiotic susceptibility (E17 highly resistant to CLR, and 23 susceptible to MNZ, CLR and AMX) (Figure 2). After 24 h of incubation, the antibacterial effects of GC012 against all the strains, increased significantly in accordance with drug concentrations ranging from ½ to 2× MIC. Total death was detected after 48 h of incubation at 2× MIC treatment. At the MIC concentration, GC012 was very effective on all strains, causing a 2–3 log_10_ CFU decrease within 24 h. As previously reported [31], DHA was active at MIC value on both the reference strain and E17 (2 µg/mL). The same results were also obtained with GC012 at lower concentration (MIC 0.125 µg/mL for ATCC 43504; 0.06 µg/mL for strains 23 and E17). Because the strain 23 has not previously been investigated, the time kill curve of DHA against strain 23 is shown in Appendix A.

### 2.4. Checkerboard Assay

The checkerboard titration performed against three strains with different antibiotic susceptibilities indicated that the interactions between GC012 and each of CLR, MNZ, and AMX were synergistic (Table 4).

### 2.5. Prevention of Biofilm Formation

The results showed that a 3-day incubation with sub-MIC concentrations of GC012 and DHA caused the concentration-dependent inhibitory effect on biofilm development, as demonstrated by crystal violet (CV) staining (Figure 3). The ½× MIC of both compounds was used because the planktonic growth of *H. pylori* was not markedly affected at this concentration.

The viability test performed by AlamarBlue^®^ (AB) assay showed a minimal biofilm inhibitory concentration (MBIC) value 0.03 µg/mL for GC012 and 0.25 µg/mL for DHA (Figure 4A,B). These results were also confirmed by the CFU count. GC012 and DHA caused a 2 log_10_ and 3 log_10_ CFU reduction, respectively (Figure 4C,D).

Furthermore, the fluorescence microscopy analysis performed after Live/Dead staining showed that the treatment with ¼ and ½ MIC GC012 (Figure 5B,C) and DHA (Figure 5D,E) induced inhibition of biofilm formation and killing of cells.

### 2.6. Eradication of Mature Biofilm

GC012 showed the capacity to eradicate biofilm, as shown by CV staining (Figure 6A) with a minimal biofilm eradication concentration (MBEC) value of 1 µg/mL as evaluated by the AB assay (Figure 6B). These results were also confirmed by the statistically significant reduction in the CFU count (Figure 6C). The biofilm treated with GC012 was characterized by a multitude of dead cells, as indicated by the red fluorescence (Figure 7B). A reduction in biofilm biomass was also obtained after treatment with 2 µg/mL DHA, as confirmed by the CV staining (Figure 6D) and a reduction of 1 log_10_ in the CFU count (Figure 6F), but it failed to achieve an MBEC, as demonstrated by the AB assay (Figure 6E). Furthermore, fluorescence microscopy showed many live cells, as indicated by the green fluorescence (Figure 7C).

### 2.7. Morphological and Ultra-Structural Modifications of DHA- and GC012-Treated H. pylori Cells as Visualized with TEM and SEM

In order to evaluate the effect of compounds on bacteria morphology, TEM and SEM analysis were performed after treatment with ½× MIC concentrations of both compounds. Morphological changes of *H. pylori* were also examined by SEM; TEM and SEM images at higher magnification were also compared. Untreated bacteria, used as control, showed intact cell wall and homogenous cytoplasm (Figure 8A and Figure 9A). The cells were rod-shaped with intact flagella and homogenous surfaces when examined under SEM (Figure 9D). Treatment with GC012 caused loss of cytoplasmic contents, formation of vesicles (Figure 8B), and separation of the inner from the outer membrane (Figure 9B); intact bacteria with dense cytoplasm were also present. Similar morphologic changes were caused by treatment with DHA (Figure 8C and Figure 9C); ultrastructural changes also include formation of cytoplasm electron-dense regions. SEM images displayed the alteration of membrane morphology (Figure 9E,F).

The treatment with both compounds at MIC concentration revealed dramatic destructive effects of the outer membrane (Appendix A).

## 3. Discussion

To overcome the resistance problem, new drugs for *H. pylori* infection are urgently required. Medicinal plant and herb extracts, including extracts from *Artemisia annua* and *Artemisia douglasiana,* possess activity against *H. pylori* infection [42,43,44,45,46,47,48], and their effect could be considered an important alternative or adjuvant therapeutic approach. Previous studies [49] demonstrated that artemisinin and a series of semisynthetic analogues were effective against drug sensitive and resistant *H. pylori* strains. In this context, we investigated the anti *H. pylori* activity of GC012. The antibacterial activity of GC012 (MIC_50_ 0.25 µg/mL) was found to be four times higher than that of MNZ and CLR, antibiotics currently used for the treatment of *H. pylori* infection [9]. GC012 was more potent than DHA in terms of MIC and MBC values. Furthermore, as the human gut harbors a complex microbial community, one important aspect of an antimicrobial treatment is the identification of new antimicrobials which do not cause dysbiosis that can increase the incidence of harmful bacteria. GC012 could be considered a specific anti-*H. pylori* agent because it is shown here not to be active against bacteria and yeast that mimic, in part, some genera of intestinal microbiota. Because the antibacterial activity of GC012 and DHA did not show differences against antibiotic-susceptible and -resistant strains of *H. pylori*, we can speculate that these molecules have a different mechanism of action with respect to the standard antibiotics tested. Moreover, the two compounds exhibited no significant cytotoxicity toward GES-1 cell lines (Appendix A), suggesting that their anti-*H. pylori* activity might not be due to general toxicity. The checkerboard titration assay revealed that GC012, in combination with standard drugs, was able to decrease the MIC values of MNZ, CLR, and AMX in all the strains.

Considering that treatment failure may be also due to the degradation of antibiotic agents by gastric acid, our results showed the GC012 maintained the same activity against all the strains over the pH range examined. It has been demonstrated that *H. pylori* develops biofilm *also* on human gastric mucosa [50], and nowadays it is considered a principal virulence factor in many localized chronic infections [51].

We found that both compounds (at ½ or ¼× MIC values) inhibited *H. pylori* biofilm formation. On mature biofilm, only GC012 showed an MBEC of 1 µg/mL (corresponding to 16× MIC). DHA did not display an MBEC at the concentrations tested. Unfortunately, its toxicity on the GES-1 cells did not allow us to test higher concentration (Appendix A). The mechanism of action of artemisinin and its derivatives against *H. pylori* are still unknown. It has been reported that their interaction may provide a mechanism to induce bacterial death [52]. In our experiments, the addition of FeCl_3_ did not modify the inhibitory effect (data not shown). It is also well known that urease plays a crucial role in the pathogenesis of *H. pylori*, and compounds that inhibit urease represent a good strategy for the treatment of infection. For this reason, we performed the urease assay after *H. pylori* treatment with different compound concentrations (1× MIC, 2× MIC, 4× MIC, 8× MIC and 16× MIC) using the urease inhibitor SUAN 32 [53] as positive control. No difference was observed between untreated and treated samples (Appendix A). TEM and SEM analysis suggests that GC012 and DHA act on the bacterial cell wall by causing retraction of the inner from the outer membrane. Furthermore, the presence of electron-dense structures in the cytoplasm suggests that it could be another target. Similar morphological changes were reported by Goswami et al. [49] after treatment with another artemisinin derivative. Thus, further studies are required to identify the specific target of artemisinin and its derivatives in *H. pylori*. The use of GC012 could be an alternative or combined therapeutic approach for treating *H. pylori* infection in planktonic and sessile forms. To the best of our knowledge, this is the first report demonstrating the activity of an artemisinin derivative on *H. pylori* biofilm.

## 4. Materials and Methods

### 4.1. Bacterial Strains Culture and Cell Culture

The antimicrobial activity of GC012 was evaluated on 24 *H. pylori* strains with different antimicrobial susceptibility, obtained from local clinical laboratories. The isolated strains have been previously used in other research studies [38,54]. Here, we used only data obtained during routine activity of our laboratory. Therefore, neither ethical approval nor patient consensus was considered necessary. The strains were identified on the basis of the colony appearance, Gram staining, and positive reactions in biochemical tests (catalase, urease, and oxidase). All strains were stored at −80 °C in Wilkins Chalgren Broth (Oxoid LTD, Basingstoke, Hampshire, England) with 20% of glycerol until their use. A reference strain of *H. pylori* (ATCC 43504) was used as control. The strains were plated on Columbia agar base (Oxoid) supplemented with 10% horse serum (HS), 0.25% bacto yeast extract (Oxoid), and incubated for 72 h at 37 °C in an microaerophilic conditions.

Gram-negative (Gram-) strains *(Escherichia coli* ATCC 25922, *Klebsiella pneumoniae* clinical strain 80, *Pseudomonas aeruginosa* ATCC 27853, *Salmonella enterica* serotype Typhimurium ATCC 14028, *Acinetobacter baumanii* clinical strain 1/F1) and Gram-positive (Gram+) strains (*Staphylococcus epidermidis* clinical strain AS, *Staphylococcus aureus* ATCC 29213, *Streptococcus mutans* clinical strain 6S8, *Enterococcus faecalis* ATCC 29212, *Lactobacillus acidophilus* ATCC 4357, *Lactobacillus rhamnosus* ATCC 53103, *Lactobacillus casei* clinical strain 40), and yeasts (*Candida albicans* ATCC 90028 and *Candida krusei* ATCC 6258) were selected for this study. Gram- strains were cultured on MacConkey Agar (Oxoid) for 18 h at 37 °C, except for *P. aeruginosa,* where agar cetrimide was used. *S. epidermidis* and *S. aureus* were cultivated on mannitol salt agar (MSA) agar (Oxoid) for 24 h at 37 °C; *S. mutans* on Columbia agar with 5% sheep blood for 48 h, and *E. faecalis* on Enterococcus agar at 37 °C for 18 h. *Lactobacillus* spp. was cultured at 37 °C on De Man, Rogosa, Sharpe Agar (MRSA) (Oxoid) for 24 h under microaerophilic conditions.

*Candida* spp. was cultured onto Sabouraud dextrose agar (SAB) (Oxoid) for 24 h at 37 °C.

Human gastric epithelial SV40-immortalized non-tumorigenic GES-1 cells (kindly provided by Prof. Dawid Kidane, Department of Pharmacology and Toxicology, University of Texas, Austin, TX, USA) were cultured in RPMI medium (HyClone, Logan, UT, USA) supplemented with 10% decomplemented fetal calf serum (FCS) (HyClone), 1% levoglutamine (Sigma-Aldrich s.r.l., Milan, Italy), 1% pen/strep (HyClone), and 10 mM Hepes (Sigma-Aldrich). Cells were maintained in a 5% CO_2_ atmosphere.

### 4.2. Reagents

DHA was obtained from the Kunming Pharmaceutical Corp. (Kunming City, China) or from Haphacen, Hanoi College of Pharmacy (Hanoi, Vietnam), and GC012 was prepared as previously described [39]; the latter was shown to be ≥98% pure by the HPLC method, as previously described [35]. Solutions of DHA and GC012 were freshly prepared in DMSO. MNZ, CLR, AMX, CPX, MCZ and ECZ purchased from Sigma-Aldrich, were dissolved and diluted according to Clinical and Laboratory Standards Institute CLSI guidelines [55].

### 4.3. Minimum Inhibitory Concentration (MIC) and Minimum Bactericidal Concentration (MBC) Determination

For *H. pylori*, the MIC was determined by microdilution method using Mega Cell^TM^ RPMI-1640 medium (Sigma-Aldrich) as previously described [56]. Briefly, about 5 × 10^5^ CFU/well were inoculated in 96-well microtiter plates containing serial two-fold broth dilution of GC012, and incubated under microaerophilic conditions (10% CO_2_ in a gas incubator). After 72 h, the lowest concentration of GC012 inhibiting visible growth was considered to be MIC. For MBC, aliquots (10 μL) of suspensions without visible growth were plated on Columbia agar base supplemented with 10% horse serum, 0.25% bacto yeast extract, and incubated at 37 °C for 72 h in microaerophily. The MBC was defined as the lowest concentration of drug, that killed ≥ 99.9% of initial inoculum. In order to verify the acid stability of GC012, MICs and MBCs at pH 2, 5 and 7 were also determined. Test compounds were diluted at working solution, treated with 0.1 N HCl to adjust the pH to 2 and 5 values, and incubated for 2 h at room temperature. Solutions prepared at pH 7 were used as controls. After incubation, MICs and MBCs determination of the acid-treated samples were performed according to the procedure described above.

MICs for *E. coli*, *K. pneumoniae*, *P. aeruginosa*, *Salmonella enterica* serotype Typhimurium, *S. epidermidis*, *S. aureus*, and *A. baumanii* were performed using a microdilution method in cation-adjusted Mueller Hinton broth (CAMHB) [55]; for S. *mutans*, CAMHB + 10% FCS was used. The MICs of *Lactobacillus spp.* were evaluated using a broth microdilution method in Wilkins-Chalgren broth (Oxoid) on 96-wells microtiter plates containing 1 × 10^5^ CFU/well. All the plates were incubated at 37 °C for 24 h in the presence or absence of 5% CO_2_, depending on the strain. For Quality Control, *E. coli* ATCC 25922 and *S. aureus* ATCC 29213 were used.

Susceptibility testing for *Candida* spp. strains was performed via a microdilution method in RPMI following the Clinical and Laboratory Standards Institute (CLSI) guidelines [57]. Strains were stored in growth broth supplemented with 20% glycerol at −80 °C until use.

### 4.4. Determination of Killing Kinetics of GC012

The bactericidal activity was evaluated using time–kill curves on *H. pylori* ATCC 43504 (resistant to MNZ), a clinical isolate E17 (highly resistant to CLR), and a clinical isolate 23 (susceptible to MNZ, CLR and AMX), using GC012 at ½×, 1× and 2× MIC concentration. Growth of bacteria in liquid medium was performed in 96-well plates in MegaCell^TM^ RPMI-1640 (Sigma-Aldrich) supplemented with 3% FCS. After 24 h and 48 h of incubation, the number of CFUs was assessed via serial dilution. The rate and extent of killing were expressed as viable count (log_10_ CFU/mL) against time.

### 4.5. Combination Effect of Antibiotics with GC012

The effects of combination of each of the standard drugs MNZ, CLR, and AMX with GC012, were determined by a checkerboard assay on clinical strains E17, 23, and the reference strain ATCC 43504, using fractional inhibitory concentration index (FICI) as previously described [58]. The inoculum size and the culture conditions were the same as those used for MIC determination. The FICI was calculated from the fractional inhibitory concentration (FIC) values of test compounds and antibiotics. The FICI ≤ 0.5, 0.5–1, 1–4, and >4.0, defined synergistic, additive, neutral, and antagonistic effect, respectively.

### 4.6. Antibiofilm Assay

To investigate whether GC012 could inhibit *H. pylori* biofilm formation, the clinical strain Hp 23, a strong biofilm producer, was used. A bacterial suspension was prepared in Brucella broth (BB) supplemented with 2% fetal calf serum, 0,3% glucose, as previously described [59], and inoculated into 24-well polystyrene microtiter plates in the absence (control) or in presence of different concentrations of GC012 and DHA (corresponding to ½×, ¼×, and ⅛× MIC value), for 72 h under static incubation. Free medium was used as blank control. Growth control (0 µg/mL) and blank control (broth + compounds) were also included.

For analysis of the eradication potential against established biofilms, the same *H. pylori* strain was inoculated into 24-well polystyrene microtiter plates. After 72 h of static incubation, the supernatant containing planktonic cells was removed, the plates were washed with 0.9% saline, and medium containing GC012 and DHA at different concentrations (1×, 2×, 4×, 8×, and 16× MIC) was added. The activity of compounds in terms of inhibition or eradication of biofilm was evaluated by crystal violet (CV) staining, alamarBlue^®^ Cell Viability Assay (AB) (Thermo Fisher Scientific, Waltham, MA, USA), colony-forming unit (CFU) count, and fluorescence microscopy analysis. Each assay was performed in triplicate at least three times.

### 4.7. Cristal Violet (CV) Staining

After incubation, the plates were washed with PBS, air dried, and stained with 0.1% CV for 15 min. The samples were rinsed with PBS and then air dried for 30 min. The dye associated with the biofilms was dissolved with 1 mL of 33% acetic acid, and 200 μL was used to measure the absorbance at 590 nm [60] with a microplate reader (Synergy IV, Biotek Instruments, Winooski, VT, USA) to determine the amount of biofilm formation.

### 4.8. Determination of Minimal Biofilm Inhibitory Concentration (MBIC) and Minimal Biofilm Eradication Concentration (MBEC) Biofilm

The antibiofilm activity was determined by the assay using the oxidation-reduction indicator Alamar Blue (AB) according to the manufacturer’s instructions. Briefly, after the drug treatment, the AB reagent was added in each well. The plates were incubated at 37 °C for 4 h in microaerophily and 200 µL used for the absorbance reading. Growth control (cells + broth + AB), media control (only broth), and negative control (broth + AB), were also included. Growth was indicated by a change in color from dark blue to pink. The percent reduction of AB in the treated and untreated samples was calculated using the formula indicated by the manufacturer. The minimum biofilm inhibitory concentration (MBIC) and minimal biofilm eradication concentration (MBEC) were defined as the lowest drug concentration, resulting in ≤50% reduction of AB and a purple/blue well, 4 h after the addition of AB, as previously demonstrated for other microorganisms [61]. Moreover, 100 µL of the bacterial solution was used for CFU count by serial dilutions and plated on Columbia agar, as described above.

### 4.9. Live/Dead Staining

The antibiofilm activity of GC012 and DHA was confirmed by using a Live/Dead BacLight bacterial viability kit (Life Technologies, Carlsbad, CA, USA) according to the manufacturer’s instructions, followed by fluorescence microscopy analysis, as previously demonstrated for other microorganisms [61].

### 4.10. MTT Assay

Cell toxicity was evaluated using the MTT (3-(4,5-dimethyl-2-thiazolyl)-2,5-diphenyl-2-H-tetrazoliumbromide) assay, as described [62]. Briefly, GES-1 cells were seeded in a 96-well microtiter plates at 10^5^ cell/mL, allowed to adhere overnight, and then treated with serial dilution of test compounds for additional 24 h, using DMSO (Sigma-Aldrich) as control. The final concentration of DMSO never exceeded 0.5%, which is not toxic to GES-1 cells. At the end of treatment, 20 µL of a 5 mg/mL solution of MTT in phosphate-buffered saline was added, and incubation continued for an additional 3 h in cell culture conditions in the dark. The blue formazan crystals were dissolved using 100 mL of lysing buffer consisting of a solution of 20% (wt/vol) sodium dodecyl sulfate (Sigma-Aldrich) and 40% *N,N*-dimethylformamide (Sigma-Aldrich) in water at pH 4.7. The plates were then read on a microplate reader using a test wavelength of 550 nm and a reference wavelength of 650 nm. The optical density at 650 nm (OD 650) was subtracted from the OD 550 to eliminate nonspecific background.

### 4.11. Urease Inhibitory Effect

The urease inhibitory effect was evaluated by a colorimetric assay as previously described [63]. This method measures the amount of ammonia produced after the reaction; the ammonia concentration is proportional to urease activity in the presence or absence of the inhibitor. In a 96-well plate, a bacterial suspension of ~10^7^ CFU/mL [64] of clinical strain 23 was treated with GC012 and DHA at 1×, 2×, 4×, 8×, and 16× MIC concentration (MIC GC012 0.06 µg/mL; MIC DHA 0.5 µg/mL). Untreated samples (0 µg/mL) and the urease inhibitor SUAN (32 µg/mL) [53] -treated samples were used as positive and negative controls, respectively. After 24 h of incubation at 37 °C in microaerophilic conditions, a solution containing urea and phenol red was added and incubated for an additional 2 h under the same conditions. The amount of ammonia produced after the reaction was measured by a microplate reader at 560 nm.

### 4.12. Transmission Electron Microscopy (TEM) and Scanning Electron Microscope (SEM)

To investigate the bacterial cell morphology after treatment with GC012 and DHA, a TEM and SEM examination was carried out. Briefly, an overnight culture of *H. pylori* strain 23 (10^6^ CFU/mL) was treated with GC012 or DHA at ½× and 1× MIC concentration and incubated for 24 h. Untreated bacteria were used as control. Samples were centrifuged at 3.000× *g* for 5 min, and bacterial pellets fixed by resuspending in fixing solution (2.5% glutaraldehyde, 4% paraformaldehyde in 0.1 M sodium cacodylate buffer pH 7.4). For TEM analysis, after a further centrifugation at 3.000× *g* for 5 min, pellets were post-fixed in 1% osmium tetroxide in 0.1 M sodium cacodylate buffer (pH 7.4) for 2 h at 4 °C, rinsed and stained with a 0.5% uranyl acetate in bidistilled water overnight at 4 °C in the dark. After rinsing with bidistilled water, the tissues were dehydrated in a series of graded ethanol solutions and embedded in epoxy resin (Electron Microscopy Science, Hatfield, PA, USA). The blocks were trimmed and sectioned using a PowerTome XL cryo-ultramicrotome (RMC Boeckeler, Tucson, AZ, USA) and ultrathin 70 nm sections were collected onto copper grids. Samples were observed using a Talos L120C transmission electron microscope (Thermo Fisher Scientific) at equipped with a 4 K digital camera Ceta CMOS (Thermo Fisher Scientific).

To prepare SEM samples, pellets were post-fixed in 2% osmium tetroxide in 0.1 M sodium cacodylate buffer (pH 7.4) and dehydrated in a series of ethanol till 100% for 10 min each step. Then, ethanol was substituted with acetone in four steps: ethanol/acetone 2:1, 1:1. 1:2, 100% acetone. The bacteria were resuspended in acetone, and 10 µL placed on a round coverslip and left to dry. The coverslip was attached to a stub and sputtered with gold. Observation was performed under a FE-SEM Sigma (Zeiss) at 5 KV and 5 mm of working distance.

### 4.13. Statistical Analysis

The differences in the means of the results between untreated and treated *H. pylori* were analyzed using Student’s *t*-test. The probability value of *p* ≤ 0.05 was considered significant.

## Figures and Tables

**Figure 1 pathogens-11-00740-f001:**
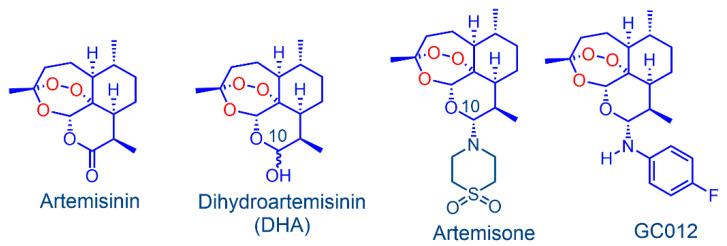
Artemisinin, and its derivatives: dihydroartemisinin (DHA) and the amino-artemisinins, artemisone and GC012.

**Figure 2 pathogens-11-00740-f002:**
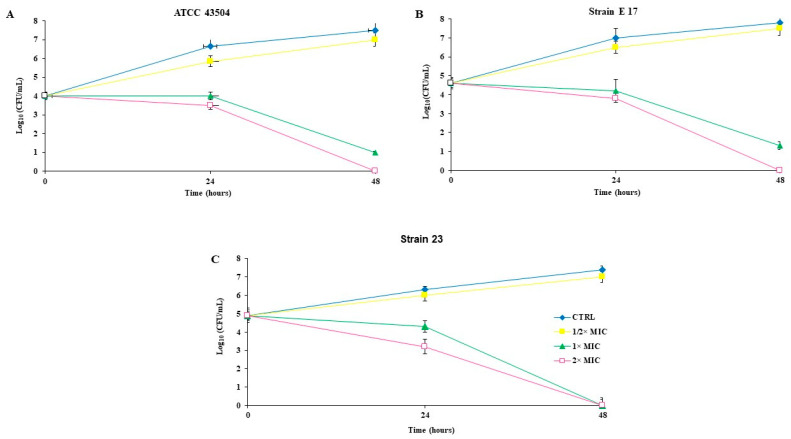
Kinetics of the killing activity of *H. pylori* ATCC 43504 (**A**), clinical isolates strain E17 (**B**), and 23 (**C**) by GC012. Antibacterial activity was evaluated in RPMI Megacell with 3% FCS in the presence or absence of the indicated drugs concentrations for different lengths of time. MIC ATCC 43504 = 0.125 µg/mL; MIC strains E17 and 23 = 0.06 µg/mL. The data are expressed as mean CFU ± SD recovered from three different experiments in triplicate.

**Figure 3 pathogens-11-00740-f003:**
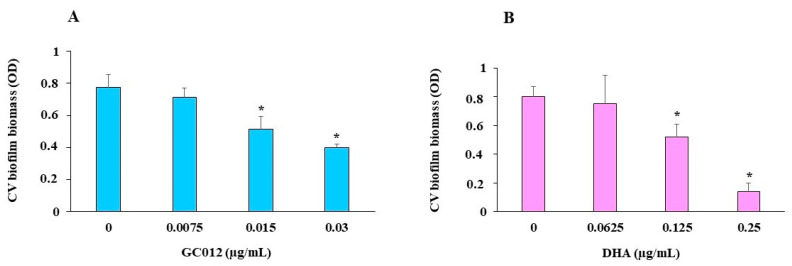
CV staining for quantification of biofilm formed by clinical strain 23 after 3 days of incubation_,_ in the presence of different GC012 (**A**) and DHA (**B**) sub-MIC concentrations, by reading OD_560._ Controls correspond to 0 µg/mL. * *p* < 0.05 vs. control.

**Figure 4 pathogens-11-00740-f004:**
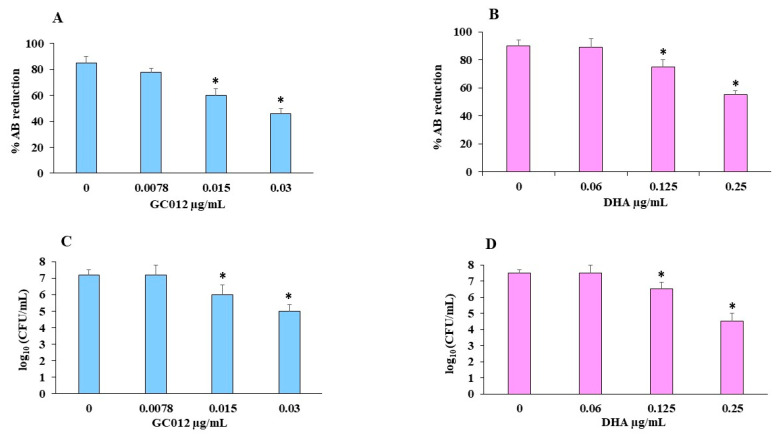
Effect of different concentration of GC012 and DHA on biofilm inhibition. (**A**) the percent reduction of AB at different GC012 concentrations compared to the corresponding untreated samples (0 µg/mL); (**B**) the percent reduction of AB at different DHA concentrations compared to the corresponding untreated samples (0); (**C**), CFU count of GC012 and (**D**) DHA-treated biofilm. * *p* < 0.05.

**Figure 5 pathogens-11-00740-f005:**
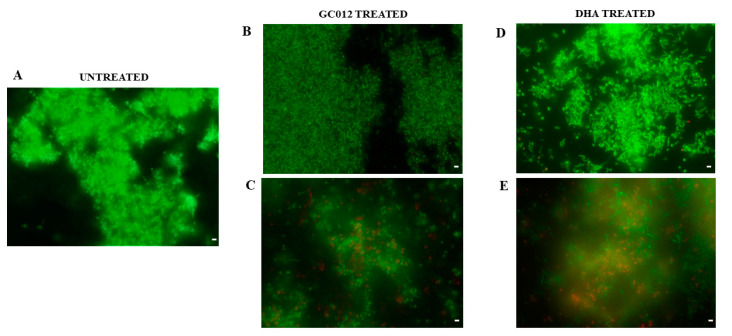
Representative *H. pylori* biofilms stained with Live/Dead kit and analyzed using fluorescence microscopy. (**A**) untreated *H. pylori* biofilm; (**B**,**D**) inhibition of *H. pylori* biofilm formation after treatment with GC012 and DHA at ¼× MIC concentration (0.015 µg/mL and 0.125 µg/mL, respectively); (**C**,**E**) inhibition of biofilm formation after treatment with GC012 and DHA at ½× MIC concentration (0.03 µg/mL and 0.25 µg/mL, respectively). The green fluorescence indicates the live cells, whereas the red fluorescence indicates the dead cells or cells with a damaged cell wall. Scale bar: 5 µm.

**Figure 6 pathogens-11-00740-f006:**
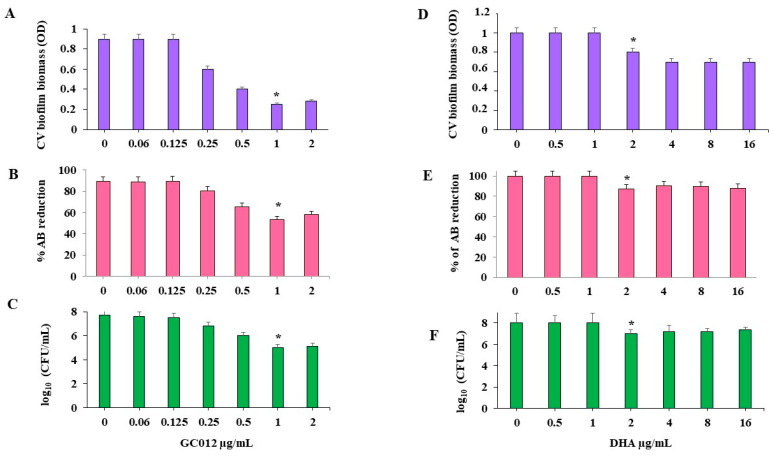
Effects of different concentrations of GC012 and DHA on 3 days mature biofilm. (**A**,**D**) CV staining; (**B**,**E**) the percent reduction of AB compared to the corresponding untreated samples (0 µg/mL); (**C**,**F**) CFU count of treated and untreated biofilm. * *p* < 0.05.

**Figure 7 pathogens-11-00740-f007:**
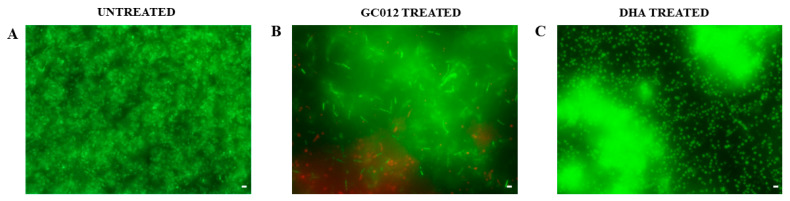
*H. pylori* biofilms stained with Live/Dead kit and analyzed using fluorescence microscopy. Representative images of (**A**) untreated *H. pylori* biofilm (clinical strain 23); (**B**) *H. pylori* biofilm after treatment with 1 µg/mL of GC012, and (**C**) *H. pylori* biofilm after treatment with 2 µg/mL of DHA. The green fluorescence indicates the live cells, whereas the red fluorescence indicates the dead cells or cells with a damaged cell wall. Scale bar: 5 µm.

**Figure 8 pathogens-11-00740-f008:**
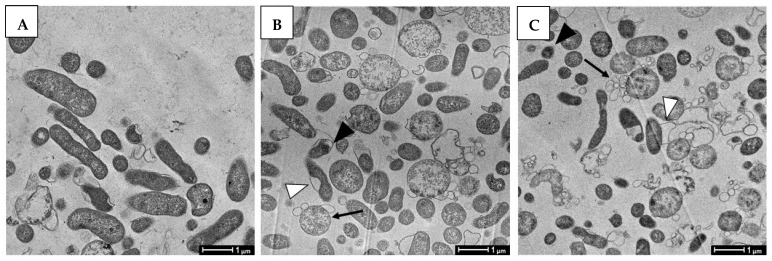
Transmission Electron Microscopy (TEM) analysis of *H. pylori* after different treatment. In all experiments, 10^6^ CFU/mL *H. pylori* were sampled after 24 h of exposure to medium (**A**); (**B**) to 0.03 µg/mL GC012; (**C**) to 0.25 µg/mL DHA. (A) Representative image that shows normal rod morphology of *H. pylori* cells with intact cell membranes (control); (B) GC012-treated cells characterized by a detachment of the inner membrane from the outer membrane (white arrowhead), vesicles (black arrows), and the formation of electron-dense structures inside the cells (black arrowhead); (C) DHA-treated cells that present comparable morphological changes. The data are representative images from two independent experiments. Scale bar: 1 µm.

**Figure 9 pathogens-11-00740-f009:**
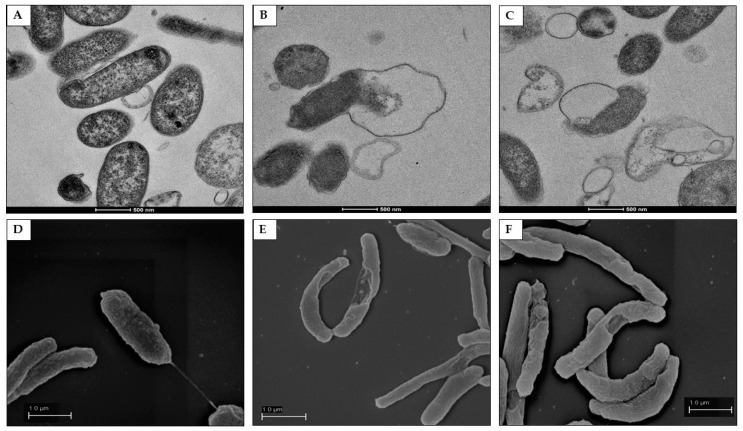
TEM (**A**–**C**) and SEM (**D**–**F**) images of *H. pylori* exposed to GC012 and DHA. (**A**,**D**) = untreated bacteria; (**B**,**E**) treated with 0.03 µg/mL GC012; (**C**,**F**) with 0.25 µg/mL DHA. (**A**–**C**), Scale bar: 500 nm; (**D**–**F**), Scale bar: 1 µm.

**Table 1 pathogens-11-00740-t001:** Anti-*H. pylori* activity of artemisinin derivatives and antibiotics assessed using a microdilution assay *.

Compounds		MIC (µg/mL)		MBC (µg/mL)	
	Range	MIC_50_	MIC_90_	Range	MBC_50_
GC012	0.03–1	0.06	0.125	0.032–0.125	0.06
DHA [38]	0.5–2	1	2	1–4	2
MNZ	1–128	1	128	0.5–128	2
CLR	0.016–256	1	256	0.032–512	0.5
AMX	0.016–0.064	0.016	0.064	0.016–0.064	0.016

* Twenty-four clinical isolates of *H. pylori,* including eight isolates highly resistant to MNZ (MIC ≥ 8 µg/mL), five resistant to CLR (MIC ≥ 0.5 µg/mL), three resistant to both MNZ and CLR, and nine MNZ, CLR and AMX susceptible, were tested. The reference strain of *H. pylori* (ATCC 43504) was used as control (EUCAST; clinical breakpoints 2021). MNZ—metronidazole; CLR—clarithromycin; AMX—amoxicillin.

**Table 2 pathogens-11-00740-t002:** Antimicrobial activity of GC012 and DHA determined by the microdilution assay.

			MIC µg/mL *		
Strains	GC012	DHA	CPX	MCZ	ECZ
		**Gram-negative bacteria**			
*E. coli* ATCC 25922	>128	>128	0.0075	n.d	n.d
*K. pneumoniae* clinical strain 80	>128	>128	n.d	n.d	n.d
*S.* Typhimurium ATCC 14028	>128	>128	n.d	n.d	n.d
*P. aeruginosa* ATCC 27853	>128	>128	n.d	n.d	n.d
*A. baumannii* clinical strain 1/F1	>128	>128	n.d	n.d	n.d
		**Gram-positive bacteria**			
*S. aureus* ATCC 29213	>128	>128	0.25	n.d	n.d
*S. epidermidis* clinical strain AS	>128	>128	n.d.	n.d	n.d
*E. faecalis* ATCC 29212	>128	>128	n.d.	n.d	n.d
*S. mutans* clinical strain 6S8	>128	>128	n.d.	n.d	n.d
*L. acidophilus* ATCC 4357	>128	>128	n.d.	n.d	n.d
*L. rhamnosus* ATCC 53103	>128	>128	n.d.	n.d	n.d
*L. casei* clinical strain 40	>128	>128	n.d.	n.d	n.d
		**Yeasts**			
*C. albicans* ATCC 90028	>128	>128	n.d	<0.125	<0.125
*C. krusei* ATCC 6258	>128	>128	n.d	<0.125	<0.125

* The MIC determination was performed by broth microdilution assay following CLSI protocols, as described in the Materials and Methods section. CPX—ciprofloxacin; MCZ—miconazole; ECZ—econazole; n.d.—not determined.

**Table 3 pathogens-11-00740-t003:** Determination of GC012 activity against *H. pylori* at different pH.

Compounds		*H. pylori* Strains MIC/MBC (µg/mL)						
	pH	ATCC 43504	102S	E34	E17	190	R5	23 (This Study)
**DHA** [38]	2	4/4	1/2	4/4	4/4	1/2	0.5/1	1/1
	5	4/4	1/2	2/4	2/4	1/2	0.5/1	0.5/1
	7	2/2	0.5/0.5	2/2	2/2	1/1	0.5/0.5	0.5/0.5
**GC012**	2	0.125/0.125	0.06/0.06	0.015/0.015	0.06/0.06	0.03/0.03	0.06/0.06	0.064/0.064
	5	0.5/0.5	0.03/0.03	0.03/0.03	0.06/0.06	0.03/0.03	0.06/0.06	0.064/0.064
	7	0.125/0.125	0.03/0.03	0.03/0.03	0.03/0.03	0.03/0.03	0.06/0.06	0.064/0.064
**MNZ**	7	128	2	1	0.064	1	128	1
**CLR**	7	0.032	0.016	0.064	256	0.032	16	0.064
Antimicrobial susceptibility		MNZ^R^ CLR^S^ AMX^S^	MNZ^S^ CLR^S^ AMX^S^	MNZ^S^ CLR^S^ AMX^S^	MNZ^S^ CLR^R^ AMX^S^	MNZ^S^ CLR^S^ AMX^S^	MNZ^R^ CLR^R^ AMX^S^	MNZ^S^ CLR^S^ AMX^S^

The treatment with GC012 at pH 2, 5, and 7 was carried out for 2 h at 25 °C. MNZ and CLR were tested only at pH 7. The MIC and MBC determination were performed via broth microdilution assay. The samples were serially diluted (two-fold) in 96-well microtiter plates containing 0.1 mL of RPMI Mega Cell with 3% FCS. MNZ^R^—metronidazole resistant; MNZ^S^—metronidazole susceptible; CLR^R^—clarithromycin resistant; CLR^S^—clarithromycin susceptible; AMX^S^—amoxicillin susceptible.

**Table 4 pathogens-11-00740-t004:** Checkerboard titration assay of GC012 in combination with antibiotics *.

	N° Isolates (%)		
	GC012		
FIC Index	CLR	MNZ	AMX
≥0.5 (synergy)	3 (100)	3 (100)	3 (100)
0.5–1 (additive)	0 (0)	0 (0)	0 (0)
1–4 (neutral)	0 (0)	0 (0)	0 (0)
>4 (antagonism)	0 (0)	0 (0)	0 (0)

* The checkerboard titration for determining the interaction effects of GC012 with standard antibiotics against *H. pylori* ATCC 43504 (resistant to MNZ), E17 (highly resistant to CLR), and 23 (susceptible to MNZ, CLR, and AMX), was performed by microdilution assay. CLR—clarithromycin; MNZ—metronidazole; AMX—amoxicillin.

## Data Availability

The datasets generated and analyzed in the current study are contained within the article or provided in the Appendix A. Details are available from the corresponding authors on reasonable request.

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
