# Peer review of "In Vitro Activity of the Arylaminoartemisinin GC012 against Helicobacter pylori and Its Effects on Biofilm"

_pathogens, 2022, doi:10.3390/pathogens11070740_

Round 1

Reviewer 1 Report

Sisto et al. examined the anti-Helicobactery pylori effect of arylaminoartemisinin, a drug derivative that is commonly used to treat malaria. The study is well designed and results are discussed properly in relation to other articles. However, few modifications are required for the article to be published- 

Line 45: remove “also”

Line 45: elaborate “MDR”

Line 46-49: Unnecessary statement. H. pylori is known to cause stomach cancer and, therefore, removal of H. pylori using antibiotics will also prevent cancer. Remove this sentence.

Line 50-52: Provide additional details…Include the components of quadruple therapy or second line eradication. Split the sentence into two separate sentences for better understanding.

Table 1: Does Table 1 indicate the data for only H. pylori strain ATCC 43504? If so, then it should be clearly mentioned in the title.

Line 210, Figure 2, and Figure 2 legend: The authors are using two different names for the same strain H. pylori ATCC 43504 and NCTC 11637 are same strain. The authors should be consistent in referring to the strain for better readership.

Figure 3: Y-axis values should be in decimals…not comma…replace comma with decimals. The authors should be consistent in their use of decimal values.

Figure 4 and 6: Be consistent in the number of decimals used in the figures.

Figure 8: Scale is very difficult to see in the figure…increase the font size

Figure 9: There is inconsistency in the scale font size…fix it

Line 465: …bismuth containing drug recommended by who? Use reference.

Line 462-477: These paragraphs do not belong to the discussion section…move these paragraphs to the introduction.

Supplementary Figure 4: Fix the Y-axis. Replace comma with decimals. Include the wavelength of the optical density.

Author Response

Reviewer 1

The Authors thank the Reviewer for all suggestions. The revisions have been included in the text.

 -  Line 45: remove “also”

It has been removed.

-  Line 45: elaborate “MDR”

MDR has been elaborated.

-  Line 46-49: Unnecessary statement. Remove this sentence.

The authors think that this aspect is important and they would like to maintain it with a modified sentence:  " So, the removal of H. pylori using antibiotics contributes to the prevention of stomach cancer in humans”. The authors wait for Reviewer final decision.

-  Line 50-52: Provide additional details…Include the components of quadruple therapy or second line eradication. Split the sentence into two separate sentences for better understanding.

The details have been added.

-  Table 1: Does Table 1 indicate the data for only H. pylori strain ATCC 43504? If so, then it should be clearly mentioned in the title.

Table 1 contains data of 24 H. pylori strains.  The strain ATCC 43504 was used as control as indicated in the legend of the table.

- Line 210, Figure 2, and Figure 2 legend: The authors are using two different names for the same strain H. pylori ATCC 43504 and NCTC 11637 are same strain. The authors should be consistent in referring to the strain for better readership.

The authors apologize for this mistake. The name has been uniformed both in the text and figures.

- Figure 3: Y-axis values should be in decimals…not comma…replace comma with decimals. The authors should be consistent in their use of decimal values.

The figure has been modified.

-Figure 4 and 6: Be consistent in the number of decimals used in the figures.

The figures have been modified.

-Figure 8: Scale is very difficult to see in the figure…increase the font size

The font size has been modified.

 -Figure 9: There is inconsistency in the scale font size…fix it

The Authors apologize for this editing mistake The images have been changed in order to have an uniform font size.

 -Line 465: …bismuth containing drug recommended by who? Use reference.

This sentence has been deleted because after moving the paragraphs to the introduction (as requested in the next point), it was a repetition.

 -Line 462-477: These paragraphs do not belong to the discussion section…move these paragraphs to the introduction.

The test has been modified as suggested.

-Supplementary Figure 4: Fix the Y-axis. Replace comma with decimals. Include the wavelength of the optical density.

The figure has been modified as suggested.

Reviewer 2 Report

I found the research article entitled "In vitro Activity of the Arylaminoartemisinin GC012 Against Helicobacter pylori and its Effects on Biofilm" very interesting and well performed. Its subject fits not only the assumptions of a Special Issue, but also the main assumptions and problems of modern therapy of pathogenic microorganisms.

Below I would like to present a short list of suggested amendments, the introduction of which may further improve the quality of the manuscript:

- Title of the Table 1: "Anti-H. pylori activity of GC012 determined by the microdilution assay*" -> Anti-H. pylori activity of artemisin derivatives and antibiotics assessed by the microdilution assay*

- "... In fact, in the mucus layer the pH increases to ~ 6, reaching a value of ~7 in the proximity of the gastric epithelium where H. pylori establishes the infection" -> please add a reference

- Table 4: please change FIC indexes as to my knowledge it should be a followed: ≤ 0.5 (synergy), 0.5 - 1.0 (additive), 1.0 - 4.0 (neutral), > 4 (antagonism)

- Titles of Figure 4 and 6: I can see some color of the text, please make it clear

- "Io order to evaluate ..." (line 365) -> In order to evaluate ...

- "... a high frequency of mutation and genetic recombination" (lines 470-471) -> ... a high frequency of mutations and genetic recombinations

- Salmonella typhimurium (line 540) -> Salmonella Typhimurium

- "... Lactobacillus casei clinical strain 40, and yeast" -> ... Lactobacillus casei clinical strain 40), and yeasts

- Section 4.3. please add info about culture medium used for H. pylori

- Lines 610 - 612: please change FIC indexes as indicated above

- "Moreover, 100 µl of reaction were used" -> Moreover, 100 µl of the bacterial solution was used

Author Response

Reviewer 2

The Authors thank the Reviewer for all suggestions. The revisions have been included in the text.

- Title of the Table 1: “Anti-H. pylori activity of GC012 determined by the microdilution assay*” -> Anti-H. pylori activity of artemisin derivatives and antibiotics assessed by the microdilution assay

The title has been modified as suggested.

-  "... In fact, in the mucus layer the pH increases to ~ 6, reaching a value of ~7 in the proximity of the gastric epithelium where H. pylori establishes the infection" -> please add a reference
The reference has been added.

- Table 4: please change FIC indexes as to my knowledge it should be a followed: ≤ 0.5 (synergy), 0.5 - 1.0 (additive), 1.0 - 4.0 (neutral), > 4 (antagonism)

The range of FICI has been modified.

-  "Io order to evaluate ..." (line 365) -> In order to evaluate ...

The mistake has been corrected

- "... a high frequency of mutation and genetic recombination" (lines 470-471) -> ... a high frequency of mutations and genetic recombinations

The text has been corrected as suggested.

Salmonella typhimurium (line 540) -> Salmonella Typhimurium

The name has been corrected.

- “… Lactobacillus casei clinical strain 40, and yeast” -> … Lactobacillus casei clinical strain 40), and yeasts

The text has been corrected.

 -Section 4.3. please add info about culture medium used for H. pylori

More informations have been added.

-Lines 610 - 612: please change FIC indexes as indicated above

FIC indexes have been changed.

-Moreover, 100 µl of reaction were used" -> Moreover, 100 µl of the bacterial solution was used

The text has been modified as suggested.
